# Production and Processing of the Radionuclide $^{76}$Br

**Karsten Franke** [1,2,*], **Jann Schöngart** [1] and **Alexander Mansel** [1,†]

1   Helmholtz-Zentrum Dresden-Rossendorf, Institute of Resource Ecology, Research Site Leipzig, 04318 Leipzig, Germany; j.schoengart@hzdr.de (J.S.)
2   Helmholtz-Zentrum Dresden-Rossendorf, Institute of Radiopharmaceutical Cancer Research, Research Site Leipzig, 04318 Leipzig, Germany
*   Correspondence: k.franke@hzdr.de
†   Deceased author.

**Abstract:** Four-dimensional visualization, i.e., three-dimensional space plus time, of fluid flow and its interactions in geological materials using positron emission tomography (PET) requires suitable radiotracers that exhibit the desired physicochemical interactions. $^{76}$Br is a likely candidate as a conservative tracer in these studies. [$^{76}$Se]CoSe was produced and used as the target material for the production of $^{76}$Br via the (p,n) reaction at a Cyclone 18/9 cyclotron. $^{76}$Br was separated from the target by thermochromatographic distillation using a semi-automated system, combining a quartz glass apparatus with a synthesis module. $^{76}$Br was successfully produced at the cyclotron with a physical yield of 72 MBq/µAh (EOB). The total radiochemical yield of $^{76}$Br from the irradiated [$^{76}$Se]CoSe target (EOS) was 68.6%. A total of 40 MBq–100 MBq n.c.a. $^{76}$Br were routinely prepared for PET experiments in 3 mL 20 mM Cl$^-$ solution. The spatial resolution of a PET scan with $^{76}$Br in geological materials was determined to be about 5 mm. The established procedure enables the routine investigation of hydrodynamics by PET techniques in geological materials that strongly sorb commonly used PET tracers such as $^{18}$F.

**Keywords:** $^{76}$Br; cyclotron; targetry; target processing; PET; phantom





## 1. Introduction

### 1.1. Motivation

The 4D visualization (three-dimensional space and time) of fluid flow and its reactions in geological materials via positron emission tomography (PET) is a key tool for a wide range of environmental transport studies [1]. The method has been successfully applied to characterize advective flow in fractures [2,3] and porous media [4–6] as well as diffusive flux in clay material [7]. Even comprehensive studies of transport processes in soil and similar surface materials have been conducted utilizing positron emitting tracers [8,9]. Depending on the scope of the study, conservative or reactive tracers are used. Conservative tracers do not interact with the surrounding geological material; they are inert [10–14]. The positron-emitting halide $^{18}$F is widely used as conservative tracer in geochemical studies, but the chemical inertness of the tracers depends strongly on the boundary conditions and the studied substrate. $^{18}$F, which is commonly used as [$^{18}$F]KF, cannot be considered a conservative tracer in the presence of strongly sorbing minerals like goethite or kaolinite [15–18] or in carbonatic materials [19]. While $^{124}$I has been previously employed in positron-tomographic transport studies [1], it is not suited for conservative transport studies in all substrates. In complex geomaterials like soils, iodine undergoes redox chemistry altering its mobility [20]. This necessitates the need for an alternative radionuclide such as $^{76}$Br as a conservative tracer in geochemical studies. To our knowledge, no studies using $^{76}$Br as a PET tracer in geochemical investigations have been conducted.

Recent publications discuss $^{76}$Br as a suitable radio tracer for theranostic radiopharamaceuticals [21–23].

*1.2. $^{76}$Br*

$^{76}$Br has a physical half-life of 16.2 h and decays via positron emission (55%) and electron capture (45%) to $^{76}$Se (stable) [24]. The decay parameters of $^{76}$Br (Table 1) do not indicate $^{76}$Br to be an ideal PET tracer. The high $\beta^+$ energies (max. > 3 MeV, mean 1.18 MeV) limit the spatial resolution to several millimeters. Pair production ($\gamma_3$–$\gamma_6$) and $\gamma$ emission close to 511 keV ($\gamma_1$) will contribute to unwanted random coincidences.

**Table 1.** End-point energies of the main $\beta^+$ and $\gamma$ from $^{76}$Br decay [24].

| | End-Point Energy [keV] | Intensity [%] | | Energy [keV] | Intensity [%] |
|---|---|---|---|---|---|
| $\beta_1$ | 871 | 6.3 | $\gamma_1$ | 559.09 | 74.0 |
| $\beta_2$ | 990 | 5.2 | $\gamma_2$ | 657.02 | 15.9 |
| $\beta_3$ | 3382 | 25.8 | $\gamma_3$ | 1216.10 | 8.8 |
| $\beta_4$ | 3941 | 6.0 | $\gamma_4$ | 1853.67 | 14.7 |
| | | | $\gamma_5$ | 2950.53 | 7.4 |
| | | | $\gamma_6$ | 2792.69 | 5.6 |

*1.3. Cross Section and Targetry*

$^{76}$Se(p,n)$^{76}$Br, $^{76}$Se(d,2n)$^{76}$Br, $^{75}$As($^3$He,2n)$^{76}$Se, and $^{75}$As($^4$He,3n)$^{76}$Se are examples of cyclotron-based pathways for the production of $^{76}$Br [25–29]. Of these, only $^{76}$Se(p,n)$^{76}$Br is a possible reaction within the specification of the cyclotron used (iba Cyclone 18/9). The energy-dependent cross section for the reaction is shown in Figure 1. The maximum of the cross section is obtained at a proton energy of $E_p \sim 13$ MeV.

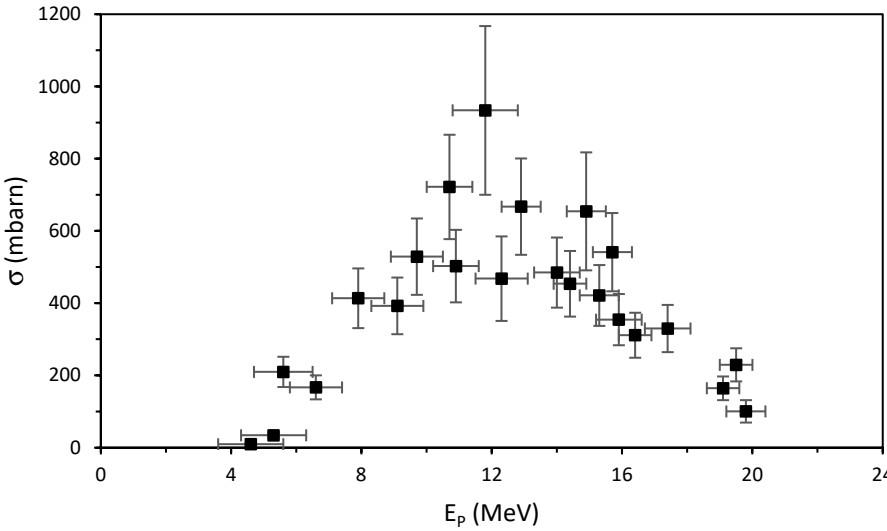

**Figure 1.** Experimental cross section $\sigma$ as function of proton energy $E_p$ for the $^{76}$Se(p,n)$^{76}$Br reaction [25].

The melting point of elemental selenium of 221 °C together with its thermal conductivity of 0.52 $Wm^{-1}K^{-1}$ [30] does not favor the use of elemental selenium as target material. Alternatively, other substances like [$^{76}$Se]ZnSe, [$^{76}$Se]SnSe, [$^{76}$Se]Cu$_2$Se, [$^{76}$Se]CuSe, [$^{76}$Se]NiSe and [$^{76}$Se]CoSe are discussed as target materials [21,31–34]. In this work, [$^{76}$Se]CoSe is used as the target material due to its thermal stability, high Se content and a low amount of unwanted secondary activation products.

## 2. Materials and Methods

*2.1. Target Material*

Elemental cobalt (powder, purity 99.998%, Alfa Aesar, Kandel, Germany) and elemental $^{76}$Se (powder, isotopic enriched 99.8 ± 0.1 atom%, STB Isotope Germany GmbH,

Hamburg, Germany) were mixed stoichiometrically and heated up to 1200 °C in an evacuated quartz glass ampule and kept at this temperature for 60 min. Afterwards the ampule was cooled down to room temperature immediately. The formed [$^{76}$Se]CoSe was removed from the ampule and pressed under argon atmosphere at 1075 °C in a cavity (Ø = 12 mm, h = 1 mm) of a niobium disc (Ø = 24 mm, h = 2 mm) (Nb foil, thickness 2 mm, purity 99.8%, Alfa Aesar, Kandel, Germany).

### 2.2. Target Irradiation

A cyclotron Cyclone 18/9 (IBA, Ottignies-Louvain-la-Neuve, Belgium) was used for the irradiation of the target. The cyclotron provided protons with an energy of 18 MeV and a current of 100 μA and deuterons with an energy of 9 MeV and a current of 40 μA on target. The target was positioned at port 4 at the Compact Solid Target Irradiation System (COSTIS, IBA Nirta target, Belgium). The target was cooled at the front with helium (60 L·min$^{-1}$) and at the back with water (16 L·min$^{-1}$). The required proton energy of 13 MeV was achieved by proper selection of the vacuum window (800 μm aluminum). Max. current on the target was 5 μA, irradiation time was from 10 to 25 min.

### 2.3. Target Processing

Thermochromatographic distillation was used after end of bombardment (EOB) to separate $^{76}$Br from the [$^{76}$Se]CoSe target [21]. Figure 2 shows a scheme of the semi-automatic system used. It combined a quartz glass apparatus with a modular synthesis machine (SCINTOMICS GmbH, Gräfelfing, Germany) controlled by the software Variocontrol (SCINTOMICS GmbH, Gräfelfing, Germany). After EOB, the irradiated target was placed in a tube furnace within a quartz glass tube (RC1) under argon atmosphere at 1055 °C for 10 min. The released $^{76}$Br was trapped in a cooling trap (ICE). After heating, the target was immediately cooled down to room temperature. The cooling trap (ICE) was removed and the transfer line was allowed to reach room temperature. A total of 15 mL water (R1) was used to rinse the system and transfer the $^{76}$Br into trap T1 containing 5 mL of water. A syringe pump was used to transfer the combined volume of 20 mL containing $^{76}$Br to cartridge C1 (Sep-Pak Accell Plus QMA Plus Light Cartridge, Waters) for purification and concentration. The cartridge was rinsed with 3 mL water (R2) and $^{76}$Br eluted with 20mM KCl solution (R3). All the released gasses passed through a sodium thiosulfate solution (T2), trapping all remaining $^{76}$Br.

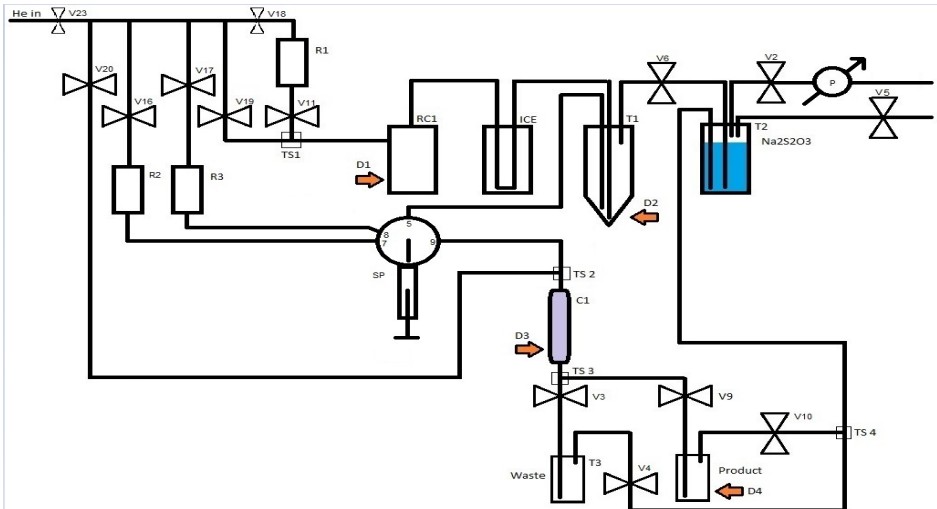

**Figure 2.** Scheme of the system used for thermochromatographic distillation (V1–V23: valves, RC1: reactor 1—quartz glass apparatus with tube furnace, ICE: cooling trap, T1: trap 1–5 mL water, SP: syringe pump, C1: QMA-cartridge, Product: product vial, T2: trap (saturated sodium thiosulfate solution), T3: waste, P: vacuum pump, R1, R2, R3: reservoirs, TS: T-connectors, D1–D4: radioactivity detectors).

### 2.4. γ-Spectrometry

Two γ-spectrometry systems were used for validation of the radionuclidic purity and quantification. To determine the produced radionuclides after EOB, the target was positioned in front of a high purity germanium detector (GEM-C5060) equipped with a Stirling cooler and DSPEC 500 (AMETEK GmbH, Meerbusch, Germany) at a distance of 575 cm. Quality and process control was carried out using a high purity germanium detector (GEM-20180-P) equipped with a Stirling cooler and DSPEC pro (AMETEK GmbH, Meerbusch, Germany) at 400 cm distance between sample and detector.

### 2.5. PET/CT

To evaluate the feasibility of $^{76}$Br as a radiotracer for positron emission tomography (PET), a phantom was measured. The phantom consists of a PTFE cylinder of 50 mm diameter with 5 drill holes (1 mm, 2 mm, 3 mm, 4 mm, and 5 mm). The bores were filled with $^{76}$Br in 20 mM KCl (as received from radiochemical workup) at an activity concentration of 31.7 MBq/mL.

PET was conducted using an 18-cassette scanner (ClearPET; Elysia-Raytest, Straubenhardt, Germany) with a cylindrical field of view of 135 mm diameter and 109 mm height. The images were reconstructed using the STIR Library [35].

Scatter correction, based on attenuation maps derived from μCT-measurements, was applied using a Monte Carlo algorithm as described by [1]. Mass attenuation coefficients for 511 keV were calculated based on data from the XCOM database [36].

A $^{22}$Na-point-source (540 Bq) was mounted on the outside of the sample. The position of this marker could be accurately identified in both in CT and PET. Image coregistration between PET and CT was achieved by using this marker as a fiducial.

## 3. Results

### 3.1. Targetry

The production of [$^{76}$Se]CoSe by mixing elemental cobalt and elemental $^{76}$Se stoichiometrically and heating up to 1200 °C in an evacuated quartz glass ampule for 60 min was tested with the $^{nat}$Se compound. We observed silvery shining amorphous deposits sticking to the quartz glass surface up to loose metallic glassy nuggets (Figures 3 and 4). This came along with grayish or reddish remains deposited at the inner surface of the quartz glass ampule indicating the formation of amorphous and polymorphous selenium species. Incomplete chemical reaction and the presence of oxygen are likely reasons. A key factor for improvement was a careful vacuum melting of the quartz glass ampule and thereby retaining the vacuum.

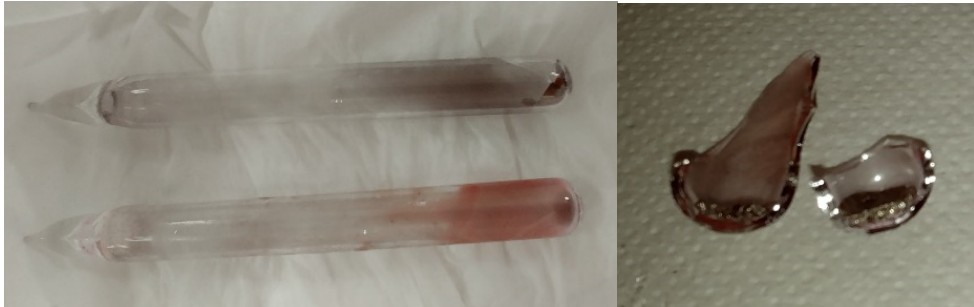

**Figure 3.** Evacuated quartz glass ampules containing CoSe after heating in the furnace at 1200 °C for 60 min. **Left**: Closed ampule—silvery shining amorphous deposits together with grayish or reddish remains at the inner surface of the quartz glass ampule. **Right**: opened ampule—porous and brittle CoSe is strongly sticking to the quartz surface.

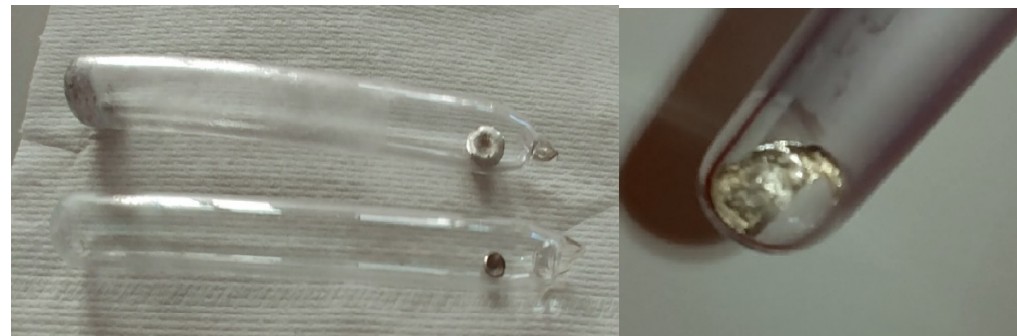

**Figure 4.** Evacuated quartz glass ampules containing CoSe after heating in the furnace at 1200 °C for 60 min. **Left**: Closed ampule—loose metallic glassy nuggets with no remains. **Right**: CoSe with metallic shine—easily removeable from quartz glass.

The formed [$^{76}$Se]CoSe was removed from the ampule and pressed under argon atmosphere at 1075 °C in a cavity (Ø = 12 mm, h = 1 mm) of a niobium disc (Ø = 24 mm, h = 2 mm) (Nb foil, thickness 2 mm, purity 99.8%, Alfa Aesar, Kandel, Germany) (Figure 5). 230 mg [$^{76}$Se]CoSe would have resulted in a ~ 26 μm homogeneous coating of the Nb cavity, assuming perfect pressing procedure. However, the cavity of the niobium disc was not entirely filled with [$^{76}$Se]CoSe after hot pressing. A more complete cover of the cavity could be achieved by an increase of [$^{76}$Se]CoSe starting material or further thorough repetitions of the pressing process. However, this would risk further loss of [$^{76}$Se]CoSe by splattering out of the cavity during the pressing procedure. We refrained from further optimization of the hot pressing, because the imperfect coverage of the cavity was neglectable for the intended $^{76}$Br production.

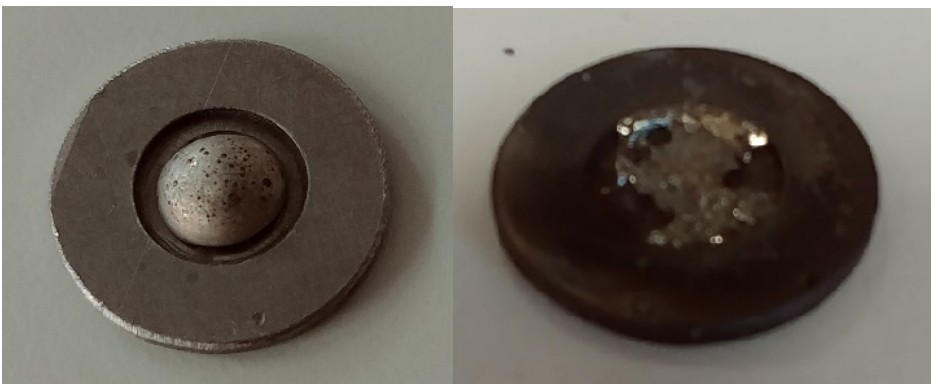

**Figure 5. Left**: Niobium disc with [$^{76}$Se]CoSe nugget on top. **Right**: Niobium disc with hot-pressed [$^{76}$Se]CoSe.

*3.2. Target Irradiation*

After irradiation, the target did not show any visible alteration. The weight of the target was controlled before and after irradiation. No weight loss was observable (*n* = 6). The deposited [$^{76}$Se]CoSe was firmly fixed on the niobium backing. After irradiation, the target was transferred to γ-spectrometry. Two radionuclides were identified (Figure 6). $^{76}$Br was produced with a yield of ~72 MBq/μAh (EOB). Aside $^{76}$Br, $^{93m}$Mo was also identified. $^{93m}$Mo is formed in the niobium target backing via $^{93}$Nb(p,n)$^{93m}$Mo reaction. The spectrum shows the prominent γ-radiation at 263.05 keV (57.4%) and 684.693 keV (99.9%).

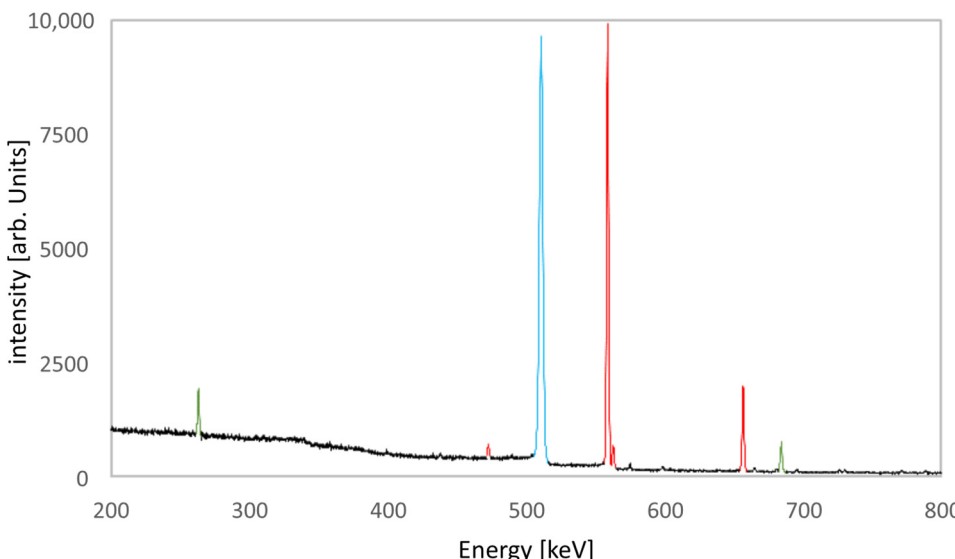

**Figure 6.** γ-spectrum section of an irradiated target disc. Identified radio nuclides are $^{76}$Br (red) and $^{93m}$Mo (green) (blue: 511 keV annihilation radiation).

### 3.3. Target Processing

The $^{76}$Br was distilled from the target at 1055 °C within 10 min and concentrated at a QMA Plus Light cartridge. Differing from published data [21,23], we used a chloride solution instead of an ammonia solution for the extraction of $^{76}$Br. The attempts to extract $^{76}$Br with ammonia solution were not successful. The switch to the chloride system also improved compatibility of the final stock solution with the required ionic strength and pH for the planned PET transport studies on geological samples.

Non-radioactive 10 μM Br$^-$ solution and 250 μM–20 mM Cl$^-$ solution were used together with ion chromatography (ICS-1600, Dionex GmbH, Idstein, Germany) to determine the minimal required Cl$^-$ concentration for the quantitative extraction of $^{76}$Br. Only the 20 mM Cl$^-$ solution allowed a quantitative extraction of $^{76}$Br. Figure 7 shows the elution of $^{76}$Br with 20 mM Cl$^-$ solution. $^{76}$Br was found in samples 3–5 of the collected 1 mL fractions. Less than 0.1% of $^{76}$Br remained at the cartridge.

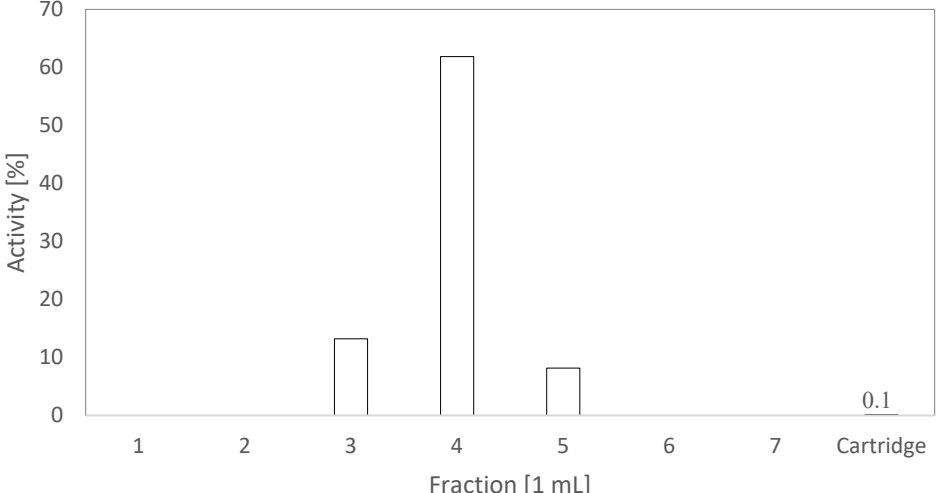

**Figure 7.** Chromatogram of the elution of $^{76}$Br from a QMA cartridge using a 20 mM chloride solution.

The overall radiochemical yield of $^{76}$Br from the irradiated [$^{76}$Se]CoSe target was about 68.6 ± 5.0% (EOS). Incomplete rinsing of the head of the quartz glass tube (RC1) after

dry distillation caused major losses of $^{76}$Br, accounting for up to 95% of the total synthesis losses. No $^{93m}$Mo was present in the final product.

### 3.4. PET

As shown in Figure 8A, the $^{76}$Br activity can be localized via PET in a PTFE sample. The maximum observed activity concentration was 2516 Bq/mL.

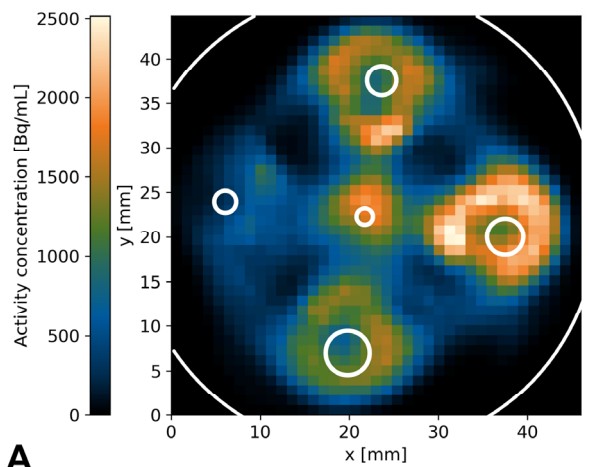
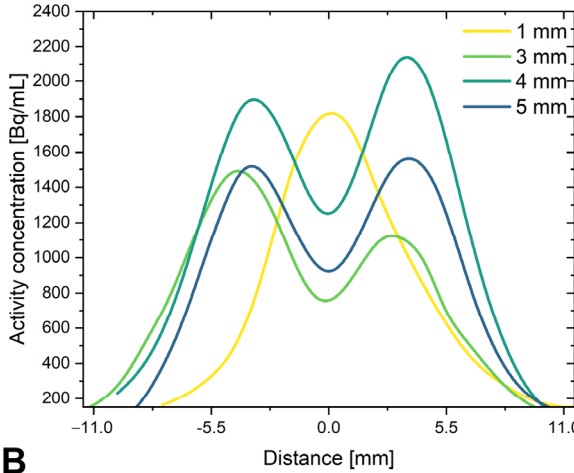

**Figure 8.** (**A**) Projected activity of the PET-phantom. The outlines of the phantom and its bore holes are outlined in white. (**B**) Line profiles across the bore holes.

Line profiles of PET activity taken across the bore positions (Figure 8B) show Gaussian equivalent FWHM of 6.4 mm (1 mm bore) to 12 mm (4 mm bore). For the bores larger than 1 mm, a local activity minimum can be observed at the position of the actual bore hole, while the peak activities occur in the surrounding material. Due to preparative error, data for the 2 mm-bore were not evaluated.

### 4. Discussion and Conclusions

The production of n.c.a. $^{76}$Br at a small 18 MeV cyclotron has been established. The target preparation via hot pressing resulted in a target disc, withstanding repeated application and showing yields comparable to published data. The target did not show alterations during irradiation; no drop of yield was observed after repeated irradiation of the target. Nevertheless, a more homogeneous distribution of target material in the cavity of the target disc could further improve the stability of the target due to a more homogeneous temperature profile within the target and also entail higher yields (MBq/µAh) in the $^{76}$Br production. The integration of a quartz glass apparatus into a modular synthesis module allows the preparation of ~100 MBq n.c.a. $^{76}$Br EOS within 3 mL. The process was optimized for extraction with Cl$^{-}$ solution. A fixed interval of 2 min was used between the 1 mL extraction steps. Further optimization in respect of the kinetics of the ion exchange within the QMA could contribute to a smaller volume and higher concentration of the final product. The radiochemical yield was 68.6 ± 5.0%. The aforementioned PET experiments on geological samples require 40–100 MBq $^{76}$Br, which fits the apparatus used here very well. The limiting factor for production of higher activities is the high amount of manual target handling required. Placement of the target in the quartz glass tube and the positioning in the furnace was done manually by hand. The current system would need optimization in respect of radio protection. Measurements on a PET phantom confirmed the feasibility of $^{76}$Br as a PET tracer. In contrast to medical applications, the high density of geomaterials limits the positron free range to reasonable values. As the line profiles in Figure 8B show, annihilation primarily happens in the polymer surrounding the bores rather than in the liquid. The achievable resolution, limited by the positron energy, is about 5 mm (cf. to

1 mm for $^{18}$F); however, much smaller features may be detected given sufficient radiotracer activity concentrations.

**Author Contributions:** Conceptualization, K.F., A.M. and J.S.; methodology, K.F., A.M. and J.S.; investigation, K.F., A.M. and J.S.; writing—original draft preparation, K.F. and J.S. K.F. and J.S. have read and agreed to the published version of the manuscript.

**Funding:** This research was funded by the Federal Ministry of Education and Research (BMBF), grant numbers 03G0900A and 02NUK066A.

**Data Availability Statement:** The original contributions presented in the study are included in the article, further inquiries can be directed to the corresponding author.

**Acknowledgments:** We thank Paul Ellison for communication and advice.

**Conflicts of Interest:** The authors declare no conflict of interest.

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
