# Peer review of "Production and Processing of the Radionuclide 76Br"

_instruments, doi:10.3390/instruments8010022_

Round 1
Reviewer 1 Report
Comments and Suggestions for Authors
Interesting improvement on the production of Br-76. The only issues I had with the manuscript was the placement of introductory materials in the methods section and vague figure descriptions (details below). I thought the limitations of the current approach were clearly discussed in the Discussion and Conclusion. Most impressive was the details of the Methods. Someone experienced in the field could easily replicate the work should they wish. Well done.
Sections 2.1 and 2.2 (up to line 64) along with table 1 and figure 1 are background information and should either be moved to introduction or discussion sections.
The data points and error bars are not described in the caption for Figure 1. Kindly add them to improve the interpretation of this image. Was this theoretical or experimental values?
Author Response
Reviewer:
Sections 2.1 and 2.2 (up to line 64) along with table 1 and figure 1 are background information and should either be moved to introduction or discussion sections.
Authors:
We moved the section 2.1 and 2.2 to the introduction
Reviewer:
The data points and error bars are not described in the caption for Figure 1. Kindly add them to improve the interpretation of this image. Was this theoretical or experimental values?
Authors:
The shown data points and error bars are taken from the given 3 references. The authors of these 3 references used the stacked-foil technique. Only Hasan et al. (2004) compared the experimental data with calculated values (ALICE – IPPE). But in Figure 1 of the manuscript only experimental data are shown.
For better readability we removed the references
Levkovski, V. N. Activation cross section nuclides of average masses (A = 40 - 100) by protons and alpha-particles with average energies (E = 10 - 50 MeV), Inter-Vesti, 1991, Moscow,
Kovács, Z., Blessing, G., Qaim, S. M., Stöcklin, G. Production of 75Br via the 76Se(p,2n)75Br reaction at a compact cyclotron. Int. J. Appl. Radiat. Isot., 1985, 36, 635-642,
and kept only
Hassan, H. E., Qaim, S. M., Shubin, Y., Azzam, A., Morsy, M., Coenen, H. H. Experimental studies and nuclear model calculations on proton-induced reactions on natSe, 76Se and 77Se with particular reference to the production of the medically interesting radionuclides 76Br and 77Br. Appl. Radiat. Isot., 2004, 60, 899-909.
We changed the figure caption to be more specific.
“Figure 1. Experimental cross section σ as function of proton energy Ep for the 76Se(p,n)76Br reaction [25]."
Reviewer 2 Report
Comments and Suggestions for Authors
The authors present an advanced methodology for the production and purification of 76Br. The methodology is an improvement over the current methods implementing a semi automated purification and importantly implementing a cooling trap to trap the liberated 76Br.
- Discussion of radioactive iodine isotopes as K[12xI] should be discussed in the introduction as alternative to 18F and 76Br as suitable isotope to be used in geological materials.
- Discussion of the chosen rinse fluid volume should be discussed. Does increasing fluid volume increase recover yield? Would letting the solution sit in the activity pool reclaim greater volumes?
- The authors note in the manuscript that the major loss of activity is due to incomplete washing of the apparatus. A detailed summary of the balance of the 76Br activity throughout the purification apparatus would be useful to the reader to determine where future improvement could be made in this process. Furthermore why was washing of the apparatus not completed as presented by Ellison.
- Can the cooling trap post activity trapped be able to be warmed to increase activity release? Would use of metal beads as a trapping source be more efficient at trapping and releasing the activity?
- Ellison et al created an updated version of their 76/77Br isolation protocol which was semi automated thermographic purification (J. Org. Chem. 2023, 88, 4, 2089–2094)
- Comment on how many irradiations were conducted without loss of material should be noted. Certain groups have seen after >10 irradiations that some loss of production yield was observed.
Comments on the Quality of English LanguageThe quality of the english is high with just small grammatical errors.
Author Response
Reviewer
Discussion of radioactive iodine isotopes as K[12xI] should be discussed in the introduction as alternative to 18F and 76Br as suitable isotope to be used in geological materials.
Authors
We added arguments in the introduction.
Reviewer
Discussion of the chosen rinse fluid volume should be discussed. Does increasing fluid volume increase recover yield? Would letting the solution sit in the activity pool reclaim greater volumes?
Authors
In our experience the volume of the rinse fluid is not a determining parameter. The 76Br sticking to the inner surfaces of the apparatus is instantaneously removed with its first contact to the rinse fluid. An increase of the volume would not contribute to higher recovery yields.
Reviewer
The authors note in the manuscript that the major loss of activity is due to incomplete washing of the apparatus. A detailed summary of the balance of the 76Br activity throughout the purification apparatus would be useful to the reader to determine where future improvement could be made in this process.
Authors
The major loss of the activity (~ 95 %) was observed at the head of the quartz glass tube (RC1). The rinse solution can not access the complete inner surface of the this part and wash of the 76Br. This is already mentioned in the manuscript:
“Incomplete rinsing of the distillation apparatus after dry distillation causes the major losses of 76Br.”
We changed the sentences to be more specific.
“Incomplete rinsing of the head of the quartz glass tube (RC1) after dry distillation causes the major losses of 76Br, accounting for up to 95% of the total synthesis losses.”
Reviewer:
Furthermore why was washing of the apparatus not completed as presented by Ellison.
Authors
We contacted Ellison to improve our recovery yield. They kindly supported us with the required information. They turned their quartz glass tube during the rinsing process upside down to improve the rinsing of the head of the quartz glass tube. Nevertheless, we did not apply this in our rinsing process, due to concerns about radioprotection. We decided to minimize the movement of this system and to avoid the lifting of the irradiated target in an elevated position.
We prefer not to discuss radioprotection issues in the manuscript. This strongly depends on the local shielding and overall setup.
Reviewer
Can the cooling trap post activity trapped be able to be warmed to increase activity release?
Authors
Indeed, we removed the transfer line from the ice bath after heating the target and brought it to room temperature.
We added the following to be more precise:
“The cooling trap (ICE) was removed and the transfer line was allowed to reach room temperature.”
Reviewer
Would use of metal beads as a trapping source be more efficient at trapping and releasing the activity?
Authors
This could be an option for further optimization. But we did observe only minor remains in this section, so we prefer to use this existing setup.
Reviewer
Ellison et al created an updated version of their 76/77Br isolation protocol which was semi automated thermographic purification (J. Org. Chem. 2023, 88, 4, 2089–2094)
Authors
Thank you for the information, we add the reference to the manuscript.
Reviewer
Comment on how many irradiations were conducted without loss of material should be noted. Certain groups have seen after >10 irradiations that some loss of production yield was observed.
Authors
The presented work summarizes the experience of 6 experiments. We add this in text to be more specific.
Round 2
Reviewer 1 Report
Comments and Suggestions for Authors
Thank you for addressing the minor points.
Reviewer 2 Report
Comments and Suggestions for Authors
Changes and answers to my corrections were satisfactory.
Comments on the Quality of English LanguageChanges and answers to my corrections were satisfactory.